# Exploring Telemedicine Usage Intention Using Technology Acceptance Model and Social Capital Theory

**DOI:** 10.3390/healthcare12131267

**Published:** 2024-06-26

**Authors:** Liang-Hsi Kung, Yu-Hua Yan, Chih-Ming Kung

**Affiliations:** 1Department of Nursing, College of Medicine, National Cheng Kung University, No. 1, University Road, Tainan City 701, Taiwan; 2u0635@mail.tmh.org.tw; 2Tainan Municipal Hospital (Managed by Show Chwan Medical Care Corporation), No. 670, Chongde Road, East District, Tainan City 701, Taiwan; 3Department of Information Technology and Communication, Shih Chien University Kaohsiung, No. 200, Daxue Road, Neimen District, Kaohsiung City 84550, Taiwan; alex@cvig.org

**Keywords:** telemedicine, technology acceptance model, urban, rural, healthcare consumers, Taiwan

## Abstract

Telemedicine technology has emerged as a pivotal solution to enhance the accessibility and efficiency of healthcare services. This study investigates the factors influencing the acceptance of telemedicine technology among healthcare professionals in Taiwan. Employing a quantitative research approach, we utilized a survey instrument adapted from the Unified Theory of Acceptance and Use of Technology (UTAUT) model. Data were collected from 325 healthcare professionals across diverse medical fields. The results indicate that perceived usefulness, ease of use, social influence, and convenience significantly influence intention to use telemedicine. Moreover, age moderates the relationship between perceived usefulness and usage intention. These findings underscore the importance of addressing both technological and social factors in promoting the adoption of telemedicine among healthcare professionals. Policy implications and recommendations for enhancing telemedicine implementation are discussed based on the study findings. Specifically, our findings highlight that perceived usefulness, ease of use, social influence, and convenience significantly impact the intention to use telemedicine technology. Age significantly moderates the relationship between perceived usefulness and usage intention. These results not only theoretically support the UTAUT model but also provide practical strategies to advance the application of telemedicine technology.

## 1. Introduction

The global outbreak of Coronavirus Disease 2019 (COVID-19) has had profound impacts worldwide, presenting significant challenges to physical health and imposing substantial burdens on public mental health [1,2,3]. In response to the pandemic’s strain on healthcare systems, telemedicine has emerged as a critical tool, transcending temporal and spatial barriers between healthcare providers and patients, thereby preventing system collapse and substituting traditional face-to-face consultations [4]. This paradigm shift has garnered unprecedented attention and is increasingly recognized as a pivotal direction in many advanced countries [5].

In Taiwan, the rapid adoption of telemedicine during the COVID-19 pandemic has highlighted challenges related to regulatory frameworks, system flexibility, and cost-effectiveness [6,7]. Overcoming these hurdles requires strategic planning to integrate telemedicine into existing medical processes while ensuring the delivery of high-quality care and fostering patient trust [8]. The severity of the COVID-19 epidemic in Taiwan prompted proactive measures, including the issuance of operational guidelines by the National Health Insurance Special Medical Institutions, expanding telemedicine services to encompass a broader segment of the population beyond their initial scope [9]. This initiative marks a significant milestone towards the widespread adoption of telemedicine in Taiwan.

The successful establishment of telemedicine services hinges on several critical factors, including institutional norms, technological advancements, healthcare providers’ readiness, and patient acceptance [10]. These factors are widely acknowledged as crucial in promoting telemedicine adoption across diverse healthcare settings [11,12]. Specifically, reference #11 emphasizes that perceived usefulness, perceived ease of use, and user attitudes significantly influence acceptance levels. Our study extends this foundational research by validating and expanding upon these findings within the specific context of telemedicine adoption in Taiwan. By doing so, we contribute novel empirical insights that enhance the theoretical foundations essential for informed policy-making and implementation strategies.

Telemedicine necessitates transformative shifts in patient–provider communication and engagement [13,14], reshaping user behavior through various applications [15]. Davis, Bagozzi, and Warshaw’s Technology Acceptance Model (TAM) provides a robust framework for understanding user behaviors towards information technology systems, emphasizing internal factors such as perceived usefulness, ease of use, attitude, and intention to use [16]. TAM is instrumental in elucidating user attitudes towards new technologies and analyzing the factors that influence their acceptance.

Social Capital Theory (SCT), proposed by Bourdieu [17] and Coleman [18], underscores the role of social networks in cultivating trust and achieving individual goals [19]. Trust, a fundamental element of social capital, facilitates cooperation and enhances organizational efficiency [20,21]. Social influence within networks significantly impacts technology adoption [22,23,24], illustrating how established trust and influence contribute to resource accumulation [25]. A comprehensive understanding of these factors is imperative for advancing telemedicine adoption and guiding future governmental policies.

This study aims to explore behavioral intentions towards telemedicine adoption using TAM and SCT as theoretical frameworks, thereby providing insights crucial for policy-making. The empirical validation of our research model further enhances our understanding of these dynamics.

## 2. Materials and Methods

### 2.1. Theoretical Frameworks and Hypotheses

In this study, the TAM and SCT were employed as theoretical frameworks to investigate factors influencing patients’ adoption of telemedicine. TAM components include perceived ease of use, perceived usefulness, and attitude, while SCT encompasses social influence and trust. By concurrently applying and integrating these theories, the study examines their combined impact on usage intention. The research hypotheses formulated are as follows (See Figure 1):

**Hypothesis** **1 (H1).***Perceived ease of use positively affects patients’ perceived usefulness of participating in telemedicine*.

**Hypothesis** **2 (H2).***Perceived usefulness has a positive effect on patients’ attitude toward telemedicine participation*.

**Hypothesis** **3 (H3).***Perceived ease of use has a positive effect on patients’ attitude toward using telemedicine participation*.

**Hypothesis** **4 (H4).***Trust has a positive effect on patients’ attitude toward participating in telemedicine*.

**Hypothesis** **5 (H5).***Social influence has a positive effect on patients’ attitude toward participating in telemedicine*.

**Hypothesis** **6 (H6).***Patients’ participation in telemedicine has a positive impact on usage intention*.

### 2.2. Sample Size Determination and Sampling

Sample size determination followed the principles recommended for structural equation modeling (SEM) studies, aiming for a minimum ratio of 10 participants per estimated parameter. Considering the study’s inclusion of 21 observed variables and a hypothesized structural model, a targeted minimum sample size of 210 participants was established.

The study population comprised patients who had previously utilized telemedicine services within [specific healthcare settings or geographic region]. A convenience sampling method was employed to recruit participants based on accessibility and willingness to participate in the study. Demographic factors such as age, gender, educational background, and marital status were considered to ensure a diverse representation.

### 2.3. Questionnaire Development and Validation

The questionnaire development process involved a comprehensive review of established scales from both the domestic and international literature to construct the research scale and measurement items. Previous studies have identified key factors influencing telemedicine adoption, including demographic characteristics and technical attributes such as perceived ease of use and usefulness [26,27], as well as the role of trust and social influence [23,28,29,30,31].

### 2.4. The Questionnaire Encompassed the following Components

Demographic Variables: This included personal attributes such as gender, age, education level, and marital status, reflecting the findings from studies by Mensah and Mi [26], Granić and Marangunić [27], and Parra et al. [32] highlighting the correlations between demographic variables and technology adoption.

Questionnaire Variables: The scale comprised six factors: perceived ease of use, perceived usefulness, trust, social influence, attitude toward using telemedicine, and usage intention. Each factor was assessed using a five-point Likert scale ranging from “strongly agree” to “strongly disagree”, ensuring a comprehensive measurement of participant perceptions and intentions [23,33,34,35,36,37,38,39].

Measurement of Outcome Variables: Outcome variables such as intention for continued usage, perceived ease of use, and others were assessed using a five-point Likert scale. These variables were treated as continuous parametric variables in our analysis. Prior to statistical analysis, we conducted normality tests and assessments of skewness and kurtosis to confirm the appropriateness of assuming a parametric distribution.

### 2.5. Handling Missing Data

Approximately 14% of collected surveys were excluded due to missing data. We considered employing missing data imputation techniques to assess the potential biases introduced by excluding incomplete responses. Our analysis indicated that missingness did not correlate systematically with key demographic variables, including educational attainment. Sensitivity analyses further confirmed that the exclusion of incomplete responses did not significantly impact the robustness of our findings.

### 2.6. Consistency in Telemedicine Platforms

All participants in our study utilized the same standardized telemedicine platform to ensure uniformity in technological interactions and minimize potential variability in association results stemming from platform differences.

### 2.7. Ethical Considerations and Data Analysis

Ethical approval for the study was obtained from the Institutional Review Board of Show Chwann Memorial Hospital (Approval No. 1101001). Data analysis commenced with descriptive statistics and reliability analysis using SPSS 21.0 to assess data quality and internal consistency. Structural Equation Modeling (SEM) was subsequently conducted using AMOS 21.0 to validate the study’s hypotheses and explore relationships among variables.

### 2.8. Software Licensing

All software used in data analysis, including SPSS 21.0 and AMOS 21.0, were licensed versions obtained through institutional agreements. The licensing details for these software applications are available upon request from the respective vendors.

## 3. Results

For the survey of 646 telemedicine patients in this institute, the gender distribution of participants was 234 (36.2%) males and 412 (63.8%) females. The majority age group was between the ages of 41 and 50 years, representing 29.7%. In terms of education, university degree holders representing 61.9% were the majority, and the minimum qualification was elementary school holders representing 0.8%. Table 1 shows the complete descriptions of the respondents.

This study used a self-response method to collect information from a single subject, which might lead to the problem of CMV. Harman’s single factor test was used [40]. The results showed that perceived usefulness, perceived ease of use, attitude, social influence, trust, and behavioral intention were six factors whose eigenvalues were >1. The explained variance of the first factor was 19.35%. It did not exceed 50% of the total explained variance (73.28%), indicating that CMV did not cause serious problems [41].

When conducting SEM analysis, it needs to include two stages: measurement and structural model analyses. First, for the measurement mode analysis stage, Bagozzi and Yi [42] proposed to adopt the following: (1) individual item reliability, (2) composite reliability (CR) of latent variables, and (3) average variance extracted (AVE) of latent variables. In the reliability part, the factor loadings (factor loading) of all variable measurement items in this research model ranged from 0.552 to 0.822, which were all >0.5 and reached statistical significance (*p* < 0.05). In the reliability analysis (reliability analysis), the Cronbach’s alpha value of each variable in this study ranged from 0.766 to 0.918, which was in line with the criteria suggested by Hair et al. [43] to an acceptable level.

In the CR part, the CR values of the variables ranged from 0.634 to 0.869, which met the criteria suggested by Fornell and Larcker [44] that CR values should be >0.6. Finally, in the mean variation extraction (AVE) part, the AVE value of each variable ranged from 0.369 to 0.624, which met the standard of AVE values suggested by Fornell and Larcker [44] that should be at least >0.3. The above results showed that the internal consistency of the questionnaire in this study is”quit’ good, with a high degree of credibility. The questionnaire dimensions are also acceptable. In the discriminant validity test, the correlation coefficient values among all aspects are significantly <1.0, which is in line with the basic assumption of discriminant validity [45] (Table 2).

Structural pattern analysis mainly verified the proposed research framework and illustrated the explanatory power of the overall pattern. In this study, AMOS 21.0 software was used for analysis, and its path coefficients were standardized to verify the four hypotheses of the research model. The results indicated that perceived ease of use has a significant positive impact on perceived usefulness (β = 0.51, *p* < 0.001); perceived usefulness has a significant positive impact on attitude toward using (β = 0.51, *p* < 0.001) = 0.37, *p* < 0.001); permitted ease of use has a significant positive effect on attitude toward using (β = 0.32, *p* < 0.001); trust has a significant positive effect on attitude toward using (β = 0.32, *p* < 0.001); social influence has no significant effect on attitude toward using (β = 0.06, *p* > 0.001); and attitude toward using has a significant positive effect on usage intention (β = 0.70, *p* < 0.001). Figure 2 illustrates the related research model path pattern.

H1–H4 and H6 were supported, but not H5. Figure 2 displays the explanatory power of each endogenous latent variable of the research model to the overall model. The path values between each dimension adopted standardized coefficients. The variation explanatory power (R^2^) of each endogenous latent variable in the research model to the overall model is as follows: the R^2^ of perceived ease of use to perceived usefulness was 26% (0.26); the R^2^ of perceived usefulness, social influence, trust, and attitude toward using was 47% (0.47); and the R^2^ of attitude toward using and usage intention was 48% (0.48).

## 4. Discussion and Implication

This study integrates the SCT and the TAM to investigate the determinants of telemedicine adoption among patients. Our results confirm that SCT and TAM effectively evaluate the variables influencing the intention to use telemedicine, with most hypotheses being substantiated. Specifically, we found significant relationships between perceived ease of use, perceived usefulness, and usage intention. However, contrary to our initial hypothesis (Hypothesis 5), social influence did not significantly impact telemedicine usage intention.

The finding that social influence was not a significant predictor contrasts with previous studies [46,47]. This discrepancy may be due to the specific context of the COVID-19 pandemic, where social interactions were limited, reducing the potential impact of social influence on telemedicine adoption.

Perceived ease of use and perceived usefulness showed a strong positive correlation, consistent with the findings of Morton et al. and Melas et al. [12,48,49]. This supports the established view that perceived ease of use indirectly affects usage intention through perceived usefulness [16,24,48,50]. Furthermore, attitude towards using telemedicine significantly influences usage intention, aligning with the results obtained by Almojaibel et al. [51], Doma et al. [52], and Shadangi et al. [53]. The more positive the attitude towards telemedicine, the higher the intention to adopt such services.

Trust emerged as a critical factor, significantly correlating with usage intention and mediating the relationship between perceived usefulness and intention. This finding aligns with van Velsen et al. [46] and Park et al. [47], highlighting trust’s role in patient acceptance of telemedicine.

## 5. Conclusions

The COVID-19 pandemic has underscored the necessity of telemedicine as a vital tool in healthcare delivery. Our study’s findings emphasize the importance of perceived ease of use, perceived usefulness, and trust in telemedicine adoption. These insights provide a comprehensive understanding of the factors influencing telemedicine acceptance and offer valuable guidance for future research and practical applications.

## 6. Recommendations

Governments and healthcare providers should collaborate to develop standardized telemedicine platforms integrated with existing medical information systems to streamline user experience and improve accessibility. Policies should be adapted to allow the broader use of telemedicine, especially for chronic disease management, reducing the need for in-person hospital visits and increasing patient satisfaction. Educating patients about the functionalities and operation of telemedicine systems is crucial. Developing user-friendly interfaces and comprehensive training materials will improve patient attitudes and increase usage intention. Initiatives to build and maintain trust in telemedicine, such as ensuring the security and efficacy of these services, are essential for enhancing patient acceptance and usage.

## 7. Limitations and Future Research Directions

While this study provides valuable insights, several limitations must be acknowledged. The self-reported nature of the questionnaire may introduce response bias, and the cross-sectional design limits causal interpretations. Future studies should adopt longitudinal designs and incorporate qualitative methods to address these limitations.

Conducting longitudinal studies will provide deeper insights into the sustained effects of telemedicine on patient outcomes and engagement. Field interviews and case studies can uncover underlying reasons for the observed lack of significant social influence, providing a richer understanding of patient attitudes. Including healthcare providers’ viewpoints can reveal additional barriers and facilitators to telemedicine adoption, offering a more comprehensive perspective. Exploring advanced technologies such as AI, machine learning, and blockchain in telemedicine can lead to innovative solutions for personalized care and enhanced data security. Comparative studies across different cultural contexts can identify unique challenges and opportunities in telemedicine adoption, informing more tailored implementation strategies.

In conclusion, this study advances our understanding of the factors influencing telemedicine adoption, providing both theoretical and practical contributions. By addressing the identified limitations and exploring future research directions, we can further enhance the efficacy and acceptance of telemedicine, ultimately improving healthcare delivery and patient outcomes.

## Figures and Tables

**Figure 1 healthcare-12-01267-f001:**
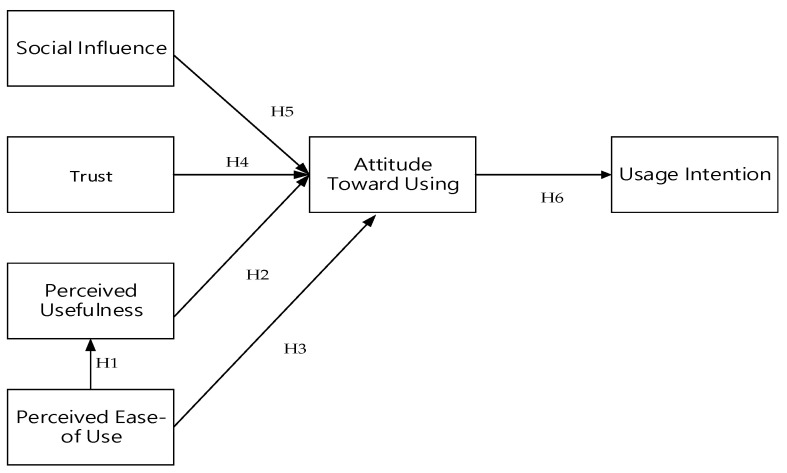
Research model.

**Figure 2 healthcare-12-01267-f002:**
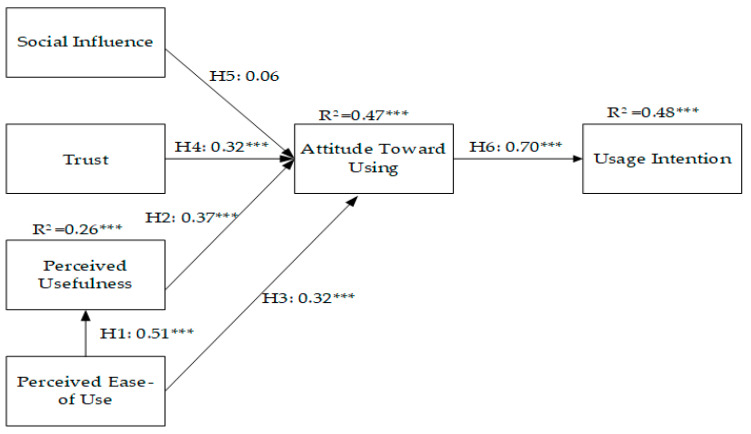
Structural model with path coefficient and R^2^ values. Note: *** indicates significance at *p* < 0.001.

**Table 1 healthcare-12-01267-t001:** Descriptions of respondents.

Measure	Item	Frequency *N* = 646	Percentage (%)
Gender	Male	234	36.2
	Female	412	63.8
Age group	<30	157	24.3
	31–40	131	20.3
	41–50	192	29.7
	51–60	127	19.7
	>61	39	6.0
Education level	Elementary school	5	0.8
	Junior high school	14	2.2
	Senior high school	133	20.6
	Bachelor’s degree	400	61.9
	Postgraduate (Master’s/Doctoral)	94	14.6
Marital status	Married	323	42.6
	Single	275	50.0
	Widowed (divorced)	48	7.4

**Table 2 healthcare-12-01267-t002:** Reliability and validity and correlation analysis.

Measure	PU	PEOU	ATU	SI	Trust	UI
Perceived Usefulness (PU)						
Perceived Ease of Use (PEOU)	0.507					
Attitude Toward Using (ATU)	0.658	0.640				
Social Influence (SI)	0.568	0.508	0.556			
Trust	0.597	0.616	0.675	0.642		
Usage Intention (UI)	0.710	0.590	0.747	0.706	0.750	
Cronbach’s α	0.766	0.867	0.780	0.918	0.879	0.927
CR	0.705	0.847	0.634	0.869	0.716	0.850
AVE	0.445	0.582	0.369	0.624	0.457	0.416
Mean	3.647	3.837	4.018	3.356	4.005	3.741
Standard Deviation	0.649	0.682	0.624	0.725	0.639	0.639

Note: CR (Composite Reliability) indicates internal consistency; values > 0.6 are acceptable. AVE (Average Variance Extracted) indicates convergent validity; values > 0.3 are acceptable.

## Data Availability

Data cannot be made publicly available owing to the fact that the privacy of individual participants cannot be compromised. However, the dataset is available from the corresponding author on reasonable request.

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
