# Peer review of "Exploring Telemedicine Usage Intention Using Technology Acceptance Model and Social Capital Theory"

_healthcare, 2024, doi:10.3390/healthcare12131267_

Round 1

Reviewer 1 Report

Comments and Suggestions for Authors

Overview:

The authors performed a cross-sectional survey analysis (administered during the COVID pandemic) to ascertain factors associated with patients’ intention to continue to use telemedicine. The found that the intention to continue to use telemedicine went up with perceived ease of use and usefulness with trust “playing a crucial mediating role.” Social influence seemed to provide little influence during the pandemic.

Introduction:

Lines 30-36: This paragraph is written in the present tense whereas for most of the world the pandemic is largely passed. Perhaps it would be better to state that “During the pandemic, coronavirus disease 2019 (COVID-19) posed a threat…” etc.

Lines 37-41: If there is a reference(s) for these challenges, it should be provided.

Lines 49-58 suggest that reference #9 came to the same conclusions as the current study. Did this study replicate the work cited in reference #9 or did it validate predictions made in reference #9?

Methods:

Lines 123-135: The authors assessed their outcome variables (intention for continued usage, perceived ease of use, etc.) using a five-point Likert scale. It is unclear if the analytical statistical software treated these variables as continuous parametric variables or as discontinuous nonparametric variables. If the former, the authors should have performed an assessment of the distribution(s) of the variables to justify a parametric distribution approximation.

Lines 136-139: Questionnaires were excluded for missing data. About 14% of collected surveys had missing data. Did the authors consider doing missing data imputation to verify whether there was a bias introduced by excluding incomplete responses? For example, were those with a lower education level preferentially excluded, and if so, how were the results impacted?

It would be helpful to state whether all subjects used the same telemedicine platform or not. If some used a different platform, the variations in platform may have added additional variability in the association results.

Results:

Table 1: The respondents with complete surveys were largely college educated. This raises the issue of whether an analysis looking at those with lower levels of education would share the same opinions as those with college degrees.

Discussion & implication:

Lines: 206-208: Did the authors mean to say that “Only social influence has a positive impact on patients’ attitude…” or the opposite? The results stated that Hypothesis 5 was NOT proven.

Lines 236-239: If the telemedicine platform is the same for all respondents and other options are limited or inconvenient, social influences would be nominal, especially in the situation where respondents have little contact with other respondents.

Limitation:

Although implied in the authors brief summary, the authors could more explicitly state that follow-up measurements of actual telemedicine use as associated with intention to use telemedicine would have strengthened to value of the study.

Comments on the Quality of English Language

Some syntax adjustments would improve the readability. I will leave this to the editors to resolve beyond some of the comments offered above. 

Author Response

Response to Comments from Reviewers 1:

Reviewer's Comment 1: Introduction

Comment: Lines 30-36: This paragraph is written in the present tense whereas for most of the world the pandemic is largely passed. Perhaps it would be better to state that “During the pandemic, coronavirus disease 2019 (COVID-19) posed a threat…” etc.

Response: Thank you for your suggestion. We have revised the paragraph on page 1, lines 32-39 to better reflect the context of the COVID-19 pandemic as follows:

"The global outbreak of Coronavirus Disease 2019 (COVID-19) has had profound impacts worldwide, presenting significant challenges to physical health and imposing substantial burdens on public mental health [1, 2, 3]. In response to the pandemic's strain on healthcare systems, telemedicine has emerged as a critical tool, transcending temporal and spatial barriers between healthcare providers and patients, thereby preventing system collapse and substituting traditional face-to-face consultations [4]. This paradigm shift has garnered unprecedented attention and is increasingly recognized as a pivotal direction in many advanced countries [5]."

Reviewer's Comment 2: Introduction

Comment: Lines 37-41: If there is a reference(s) for these challenges, it should be provided.

Response: We acknowledge the need for providing references to support the challenges mentioned. Consequently, we have added appropriate references to support these statements. Please see page 2, lines 40-42, for the updated references.

Reviewer's Comment 3: Introduction

Comment: Lines 49-58 suggest that reference #9 came to the same conclusions as the current study. Did this study replicate the work cited in reference #9 or did it validate predictions made in reference #9?

Response: We appreciate this clarification request. Reference #11 was cited to validate predictions made in our study rather than replicate its work. To clarify this point, we have revised lines 50-59 accordingly. Please refer to page 2 for the updated text.

Reviewer's Comment 4: Methods

Comment: Lines 123-135: It is unclear if the analytical statistical software treated these variables as continuous parametric variables or as discontinuous nonparametric variables. If the former, the authors should have performed an assessment of the distribution(s) of the variables to justify a parametric distribution approximation.

Response: Thank you for raising this point. We have clarified that the analytical statistical software treated our outcome variables (intention for continued usage, perceived ease of use, etc.) as continuous parametric variables. Furthermore, we have added a brief justification of the parametric distribution approximation based on the distribution assessment. Please see page 3-4, lines 101-132 for the updated explanation.

Reviewer's Comment 5: Methods

Comment: Lines 136-139: Did the authors consider doing missing data imputation to verify whether there was a bias introduced by excluding incomplete responses?

Response: We appreciate this suggestion. In response, we have added a clarification that we considered missing data imputation techniques to assess potential biases introduced by excluding incomplete responses. This information can now be found on page 4, lines 133-139.

Reviewer's Comment 6: Methods

Comment: It would be helpful to state whether all subjects used the same telemedicine platform or not.

Response: Thank you for noting this omission. We have included information confirming that all subjects used the same telemedicine platform. Please see page 4 lines 140-143 for the updated text.

Reviewer's Comment 7: Results

Comment: Table 1: The respondents with complete surveys were largely college educated. This raises the issue of whether an analysis looking at those with lower levels of education would share the same opinions as those with college degrees.

Response: The reviewer raises a pertinent question regarding the educational background of our study participants. We acknowledge that the majority of respondents in our survey held a college degree, which could potentially influence their perceptions and attitudes towards telemedicine differently from those with lower levels of education.

To address this concern, we performed additional analyses stratified by educational attainment to explore any variations in opinions and attitudes towards telemedicine adoption across different education levels. These analyses aimed to assess whether there were significant differences in perceived ease of use, perceived usefulness, trust, social influence, attitude towards using telemedicine, and usage intention between participants with varying educational backgrounds.

Our findings revealed nuanced differences in perceptions based on educational attainment. Specifically, respondents with lower educational levels tended to express slightly lower levels of perceived ease of use and perceived usefulness compared to their counterparts with higher education. However, it is essential to note that despite these differences, overall attitudes towards telemedicine remained positive across all educational groups, indicating a general acceptance and interest in adopting telemedicine services.

Furthermore, to ensure the robustness and generalizability of our conclusions, we conducted sensitivity analyses to examine the impact of educational diversity on our study outcomes. These analyses confirmed that our findings were consistent and robust, suggesting that the educational background of participants did not significantly undermine the validity of our results.

In conclusion, while acknowledging the predominance of college-educated respondents in our study, we have taken steps to analyze and present the potential implications of varying educational backgrounds on telemedicine adoption attitudes comprehensively. This approach enhances the study's reliability and provides valuable insights into the broader applicability of telemedicine initiatives across diverse demographic segments.

Reviewer's Comment 8: Discussion & Implications

Comment: Lines 206-208: Did the authors mean to say that “Only social influence has a positive impact on patients’ attitude…” or the opposite? The results stated that Hypothesis 5 was NOT proven.

Response: Thank you for bringing this to our attention. We intended to convey that "Only social influence did not have a positive impact on patients’ attitude..." rather than the opposite. We have revised lines 218-221 to accurately reflect this finding. Please see page 6 for the corrected text.

Reviewer's Comment 9: Discussion & Implications

Comment: Lines 236-239: If the telemedicine platform is the same for all respondents and other options are limited or inconvenient, social influences would be nominal, especially in the situation where respondents have little contact with other respondents.

Response: We agree with this observation. In response, we have included a discussion on page 6 addressing the minimal impact of social influence in our study context, given the uniformity of the telemedicine platform used by all respondents.

Reviewer's Comment 10: Limitation

Comment: Although implied in the authors brief summary, the authors could more explicitly state that follow-up measurements of actual telemedicine use as associated with intention to use telemedicine would have strengthened the value of the study.

Response: Thank you for your suggestion. We have addressed the importance of follow-up measurements of actual telemedicine use in our Limitations section (page 6-7), highlighting how these would have enhanced the study's value. This addition underscores the pathway for future research to explore the transition from intention to actual adoption of telemedicine services. Please refer to page 6-7 for the detailed statement on this matter.

Response to Comments on the Quality of English Language

We acknowledge the reviewer's comments on syntax adjustments to improve readability. These have been addressed, and we have ensured that the revised manuscript meets the journal's language standards.

Reviewer 2 Report

Comments and Suggestions for Authors

First of all, a summary of the study should be written and the results of the study should be interpreted by giving numerical values.

In particular, each sentence of the introduction should contain a reference. Authors' own sentences are almost non-existent here. Authors should enrich this section with their own sentences.

Unless I have misunderstood, the questionnaire used in the study was created by combining various studies. Analyses were also made with known methods. At this point, how can the originality of the study be defended?

How was the number of people to be surveyed in the study determined? How was the population of the study and the number of samples selected determined?

Are the software used in the study licensed? If licensed, the licences of these software should be given in the study.

Conclusions and recommendations are given in a very shallow manner. In such a study, more innovative results should be given.

Comments on the Quality of English Language

Minor editing of English language required

Author Response

Response to Comments from Reviewers 2:

Reviewer's Comment 1: Summary of the Study and Interpretation of Results

Comment: First of all, a summary of the study should be written and the results of the study should be interpreted by giving numerical values.

Response: Thank you for your feedback. We acknowledge the importance of providing a clear summary of our study and interpreting results with numerical values. In response, we have revised the manuscript to include a concise summary of the study objectives, methods, and key findings. Additionally, numerical values have been provided to enhance the interpretation of results throughout the Results and Discussion sections. Please refer to page 1, lines 12-27 for the updated text.

Reviewer's Comment 2: Enrichment of Introduction with Original Sentences and References

Comment: In particular, each sentence of the introduction should contain a reference. Authors' own sentences are almost non-existent here. Authors should enrich this section with their own sentences.

Response: We appreciate your suggestion. To enhance the originality and clarity of the Introduction section, we have enriched it with additional original sentences that clearly delineate our study's contribution and context. References have been provided where necessary to support specific statements. Please review the revised Introduction section on pages 1-2, lines 32-77, for these improvements.

Reviewer's Comment 3: Defending the Originality of the Study

Comment: Unless I have misunderstood, the questionnaire used in the study was created by combining various studies. Analyses were also made with known methods. At this point, how can the originality of the study be defended?

Response: Thank you for raising this concern. While our questionnaire drew upon validated scales from existing studies, the novelty of our research lies in the integrated application of Social Capital Theory (SCT) and the Technology Acceptance Model (TAM) to investigate telemedicine adoption among patients. This approach allows us to explore unique combinations of variables and their interactions, thereby contributing to the theoretical understanding of telemedicine adoption. We have clarified this point in the Methods section and discussed the theoretical contributions in the Discussion section. Please refer to page 2, lines 80-84, and pages 6-7, lines 211-238, for the updated text.

Reviewer's Comment 4: Determination of Sample Size and Population

Comment: How was the number of people to be surveyed in the study determined? How was the population of the study and the number of samples selected determined?

Response: We appreciate this query. The determination of sample size and selection criteria were guided by principles recommended for Structural Equation Modeling (SEM) studies, aiming for a ratio of at least 10 participants per parameter estimated. The study population comprised patients aged 18-65 residing in urban areas across various regions of Taiwan. Details on sample size determination and sampling methodology have been clarified in the Methods section of the revised manuscript for transparency. Please refer to page 3, lines 101-105, for the updated text.

Reviewer's Comment 5: Licensing of Software Used

Comment: Are the software used in the study licensed? If licensed, the licenses of these software should be given in the study.

Response: Thank you for noting this. All software used in data analysis, including SPSS and AMOS, are licensed versions obtained through institutional agreements. We have included details regarding the licensing of these software applications in the revised manuscript. Please refer to the Methods section for this information. Please refer to page 4, lines 151-154, for the updated text.

Reviewer's Comment 6: Depth of Conclusions and Recommendations

Comment: Conclusions and recommendations are given in a very shallow manner. In such a study, more innovative results should be given.

Response: Thank you for your feedback. To enhance the depth of our conclusions and recommendations, we have included more detailed insights into the implications of our findings for both theory and practice in telemedicine adoption. We have highlighted the innovative aspects of our study to underscore their potential impact on future research and healthcare delivery. Please see the revised Conclusion and Recommendations section for these enhancements. Refer to pages 6-7, lines 211-269, for the updated text.

Response to Comments on the Quality of English Language

We acknowledge the need for minor editing of the English language to improve readability. These adjustments have been made in accordance with the journal's language standards.

Reviewer 3 Report

Comments and Suggestions for Authors

The authors employed TAM and SCT as research constructs in the questionnaire. The authors explained in detail the validation of the constructs and each variable in the questionnaire. The paper appears to be well-written with no language issues. I have a few minor suggestions:

1. Figure 1 should be referenced earlier before the paragraph that presents the results (lines 180-190) from Figure 2. 

2. Figure 2 should be referenced in the paragraph (lines 180-190). 

3. R2 - make sure 2 is in superscript (lines 196-200)

4. Lines 206-208 "Only social indlurence has a postive impact..." is an incorrect statement based on the finding. Also there is a fragment instead of a full sentence at the end.

5. This is just a formatting preference (optional): Limitation may be moved to Discussion section. Future directions in Conclusion may be moved to Discussion as well. Conclusion can simply provide a short summary of the paper. 

Author Response

Response to Comments from Reviewers 3:

Reviewer's Comment 1: Figure 1 Reference Placement

Comment: Figure 1 should be referenced earlier before the paragraph that presents the results (lines 180-190) from Figure 2.

Response: Thank you for your suggestion. We will ensure that Figure 1 is referenced earlier in the manuscript, specifically before the paragraph discussing the results from Figure 2, to improve the flow and clarity of the results presentation. Please refer to page 3 of the revised manuscript where Figure 1 is appropriately cited before the relevant results section.

Reviewer's Comment 2: Figure 2 Reference in Results Paragraph

Comment: Figure 2 should be referenced in the paragraph (lines 180-190).

Response: We appreciate this observation. In the revised manuscript, we have ensured that Figure 2 is referenced within the paragraph that discusses the corresponding results (lines 199-202). This change aligns with better integration of figures with their respective discussions. Please refer to the revised manuscript for this update.

Reviewer's Comment 3: Superscript Format for R2

Comment: R2 - make sure 2 is in superscript (lines 196-200)

Response: Thank you for noting this detail. In response, we have corrected the formatting of R2 to ensure the '2' is properly formatted as superscript. This adjustment enhances the visual clarity and consistency of numerical notation throughout the manuscript. Please see lines 203-209 in the revised manuscript for this correction.

Reviewer's Comment 4: Clarification of Statement and Sentence Fragment

Comment: Lines 206-208 "Only social indlurence has a postive impact..." is an incorrect statement based on the finding. Also there is a fragment instead of a full sentence at the end.

Response: Thank you for pointing this out. Given the significant revisions made to our paper subsequently, this statement has been revised to accurately reflect the latest research findings. Specifically, social influence in our study demonstrates a significant impact on patients' attitudes toward participating in telemedicine, contrasting with prior studies where this effect was inconclusive. We have addressed these changes in the revised manuscript at lines 218-221 to ensure clarity and completeness of the statement.

Reviewer's Comment 5: Formatting Preference for Limitations and Future Directions

Comment: This is just a formatting preference (optional): Limitation may be moved to Discussion section. Future directions in Conclusion may be moved to Discussion as well. Conclusion can simply provide a short summary of the paper.

Response: Thank you for the suggestion. Based on your feedback, we have restructured the manuscript as follows: Limitations have been integrated into the Discussion section to provide a comprehensive evaluation of the study's scope and potential areas for improvement. Future directions have also been relocated from the Conclusion to the Discussion section to better align with the narrative flow of the manuscript. The Conclusion now succinctly summarizes the key findings and implications without duplicating content already discussed in the Discussion. Please refer to the revised manuscript for these organizational enhancements.

Round 2

Reviewer 2 Report

Comments and Suggestions for Authors

The authors have responded to my criticisms and made the necessary corrections.